# CoroFinder: A New Tool for Real Time Detection and Tracking of Coronary Arteries in Contrast-Free Cine-Angiography

**DOI:** 10.3390/jpm12030411

**Published:** 2022-03-06

**Authors:** Paolo Zaffino, Maria Francesca Spadea, Ciro Indolfi, Salvatore De Rosa

**Affiliations:** 1Department of Experimental and Clinical Medicine, Magna Graecia University, 88100 Catanzaro, Italy; p.zaffino@unicz.it (P.Z.); mfspadea@unicz.it (M.F.S.); 2Department of Medical and Surgical Sciences, Magna Graecia University, 88100 Catanzaro, Italy; 3Mediterranea Cardiocentro, Via Orazio, 2, 80122 Naples, Italy

**Keywords:** coronary angiography, tracking, geometric modeling, coronary artery calcifications

## Abstract

Coronary Angiography (CA) is the standard of reference to diagnose coronary artery disease. Yet, only a portion of the information it conveys is usually used. Quantitative Coronary Angiography (QCA) reliably contributes to improving the measurable assessment of CA. In this work, we developed a new software, CoroFinder, able to automatically identify epicardial coronary arteries and to dynamically track the vessel profile in dye-free frames. The coronary tree is automatically segmented by Frangi’s filter in the angiogram’s frames where vessels are contrasted (“template frames”). Afterward, the image similarity among each template frame and the dye-free images is scored by cross-correlation. Finally, each dye-free image is associated with the most similar template frame, resulting in an estimation of vessel contour. CoroFinder allows locating the position of coronary arteries in absence of contrast dye. The developed algorithm is robust to diverse vessel curvatures, variation of vessel widths, and the presence of stenoses. This article describes the newly developed CoroFinder algorithm and the associated software and provides an overview of its potential application in research and for translation to the clinic.

## 1. Introduction

Since 30 October 1958, when the first accidental injection of contrast dye paced the birth of coronary angiography (CA), this diagnostic technique has progressively evolved and its use widespread. Experimental development around CA has initially been mainly focused on the assessment of the coronary lumen profile, with progressive refinements in its applications. One of the first and natural evolutions of the technique has been the development of quantitative coronary angiography (QCA) that has been rapidly adopted for precise measurement of coronary arteries [1,2]. Due to its better reliability in comparison with visual assessment of CA, QCA has been adopted by many interventional cardiology laboratories worldwide and in all clinical studies concerning the use of CA [3]. A further development stemming from the latter has been more recently the three-dimensional QCA. However, CA continues to evolve, with the latest development being the use of angiographic projections to predict the impact of lumen narrowing on coronary flow by computational flow dynamics [4,5,6,7].

The measurement of coronary artery calcification (CAC) by means of Coronary Computed Tomographic Angiography (CCTA) can improve risk stratification in patients [8]. For this reason, international practice guidelines recommend the measurement of CAC and calculation of the calcium score to improve the stratification of cardiovascular risk [9]. Yet, several barriers are still present to its widespread adoption, including the availability of technology and the costs of the analysis software and protocols and training [10,11,12]. For these reasons, we intended to develop a tool to allow the location of coronary arteries in coronary angiograms without the presence of contrast dye. In fact, this represented a barrier for some clinical applications of AC, such as identification and measurement of calcium, which would not be currently possible during CA since the presence of the radiopaque contrast agent almost completely masks calcifications. Further potential clinical applications are described in the discussion section.

This article describes the development of CoroFinder, its working algorithm and the associated software, also providing an overview of its application in research and potential for translation to the clinic.

## 2. Materials and Methods

In the proposed strategy, contours of coronary arteries are automatically segmented by Frangi’s filter in the angiogram’s frames where vessels are contrasted (“template frames”) [13]. Afterward, image similarity among each template frame and dye-free frame is scored by cross-correlation. Finally, each dye-free frame is associated with the most similar template image, and contours are transferred accordingly.

Source images. Patients undergoing coronary angiography for clinical indication at the Interventional Cardiology Laboratory of the Magna Graecia University Hospital (Catanzaro, Italy), were used as exemplary use-cases for testing the proposed software. Coronary angiograms were performed during cardiac catheterization using a GE Innova 3100 Cath/Angio System (GE Healthcare, Little Chalfont, Buckinghamshire, UK). For this purpose, different projections acquired by different angles were collected. Angiographic acquisition included at least one dye-free cardiac cycle before the injection of the contrast medium (two cardiac cycles in total). The images were finally anonymized and exported in Dicom format.

Software implementation. CoroFinder was written in Python [14], on top of wxPython, Pydicom [15], SimpleITK [16], NumPy [17], and scikit-image libraries [18]. Cross operating system execution is fully supported while, to binary package the code, PyInstaller utility was used.

Algorithm testing. To test the algorithm, multiple projections from two use-case patients were used. For each projection, frames with contrast dye were manually selected. The CoroFinder algorithm was exploited to predict the coronary location for all the frames, both with and without contrast agent. Results were assessed both using image overlay and by visualization of the output generated by CoroFinder for comparison with dye-enhanced coronary angiography to verify correspondence between the coronary artery contour generated by the algorithm and the actual coronary angiogram. Visual assessment was performed by two independent and board-certified interventional cardiologists.

## 3. Results

CoroFinder software included multiple steps to allow automatic identification of epicardial coronary arteries, as well as to track them in contrast-free images. Testing of the fully automated algorithm we developed for vessel segmentation showed that the algorithm is robust to diverse vessel curvatures, variation of vessel widths, and the presence of stenoses.

### 3.1. Logical Workflow

The logical workflow was designed including the following steps: (i) selection of the frames where the coronary tree is well visible by contrast. Hence, the first frame where the contrast dye starts to be injected and the last frame before the means is washed out are identified. Even if the two time-points are automatically identified by the algorithm itself, they can be manually adjusted by the user using dedicated buttons. (ii) In step 2, vessel contour is automatically detected for each of the selected frames. Specifically, segmentations are computed by using a Frangi filter, followed by morphological operations to remove the noise. (iii) The third phase is to compute image similarity between each dye-free frame and the segmented ones corresponding to the same phase of the cardiac cycle. Finally, by taking advantage of the information computed in step 3, phase 4 provides an estimation of vessel location in dye-free frames, applying a colored contour. Figure 1 reports a schematic representation of the workflow, while Figure 2 shows the graphical interface of the main window of the software.

### 3.2. Algorithm Testing

To test the algorithm, multiple projections from two use-case patients were used. The CoroFinder algorithm was exploited to map the coronary location for all the frames, both with and without a contrast agent. The virtual contour generated by CoroFinder to estimate the position, morphology, and extension of the epicardial coronary artery showed cross-correlation with the actual angiogram at image similarity assessment. Figure 3 depicts some exemplary frames with highlighted coronary vessels, showing the comparison process. Two different colors were used to differentiate between CoroFinder-predicted (yellow contour) and segmented contours from contrast-enhanced angiograms (green contour). Cross-comparison using both image overlay and visual assessment by two independent examiners yielded 100% agreement.

## 4. Discussion

In this paper, we describe CoroFinder, a new software we developed to automatically detect and track the profile of epicardial coronary arteries in dye-free images.

CoroFinder allows locating the position of coronary arteries and defining the their profile in absence of contrast dye. This feature was not available until now and might be particularly useful for several clinical applications, such as enabling the detection of coronary calcifications and their quantification. The developed algorithm turned out to be robust to diverse vessel curvatures, variation in vessel widths, and the presence of stenoses.

The identification and measurement CAC in patients undergoing CA is particularly interesting, and it is the first we will further develop and validate for clinical use. In fact, CAC measurement might further improve risk stratification in patients undergoing diagnostic coronary angiography, with the potential to better inform decisions on treatment [9], overcoming current barriers limiting standard CT-based CAC assessment [10,11,12].

In addition, using the coronary contour generated by CoroFinder overlayed on fluoroscopy might be useful to help interventional cardiologists to better manage coronary interventions with a reduced use of contrast agents, thereby lowering the risk for contrast-induced renal damage [19]. Furthermore, a quantitative measurement of coronary calcifications might be useful to risk stratify target lesions before proceeding to percutaneous coronary intervention (PCI), to select the most appropriate PCI technique upfront. This might be helpful to select the most appropriate strategy to manage coronary calcium among the multiple technologies available nowadays [20]. Other potential applications include additional features to guide coronary interventions, such as the identification of coronary stents, their deformation, and the potential recognition of fractures or any other malfunction.

Further clinical studies are needed to assess the feasibility and to validate its clinical use for each of the possible application listed above and to assess its diagnostic performance. In this regard, some limitations should be mentioned. CoroFinder is a stand-alone application in its current form. Hence, initial feasibility studies to evaluate possible clinical application will be performed offline. In fact, the location of coronary arteries in dye-free frames is not provided real time during the performance of coronary angiography. Integration with the angiographic equipment will be technically easy to accomplish. However, authorization pathways for direct integration with commercially available coronary angiography equipment might represent a barrier to the validation of full-online use. CoroFinder does not provide a quantitative assessment of coronary lumen. In fact, this was not our focus, as commercially available angiographic equipment is already able to provide these features.

## 5. Conclusions

In conclusion, multiple clinical applications can be envisioned for CoroFinder, as it can identify coronary artery contours at angiography and to dynamically track coronary artery position in frames without contrast agents. Potential clinically useful applications include the identification and quantification of coronary calcifications. Future studies are needed to develop and to validate potential clinical applications of CoroFinder.

## Figures and Tables

**Figure 1 jpm-12-00411-f001:**
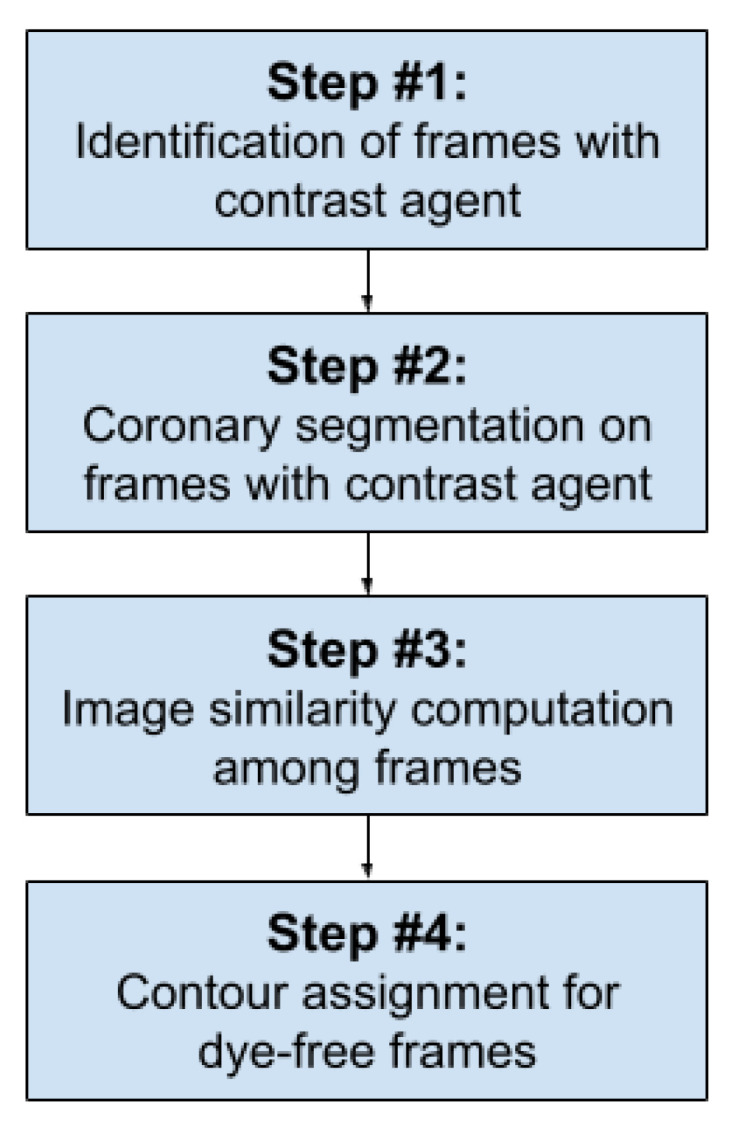
The figure shows the logical workflow behind the software’s graphical interface.

**Figure 2 jpm-12-00411-f002:**
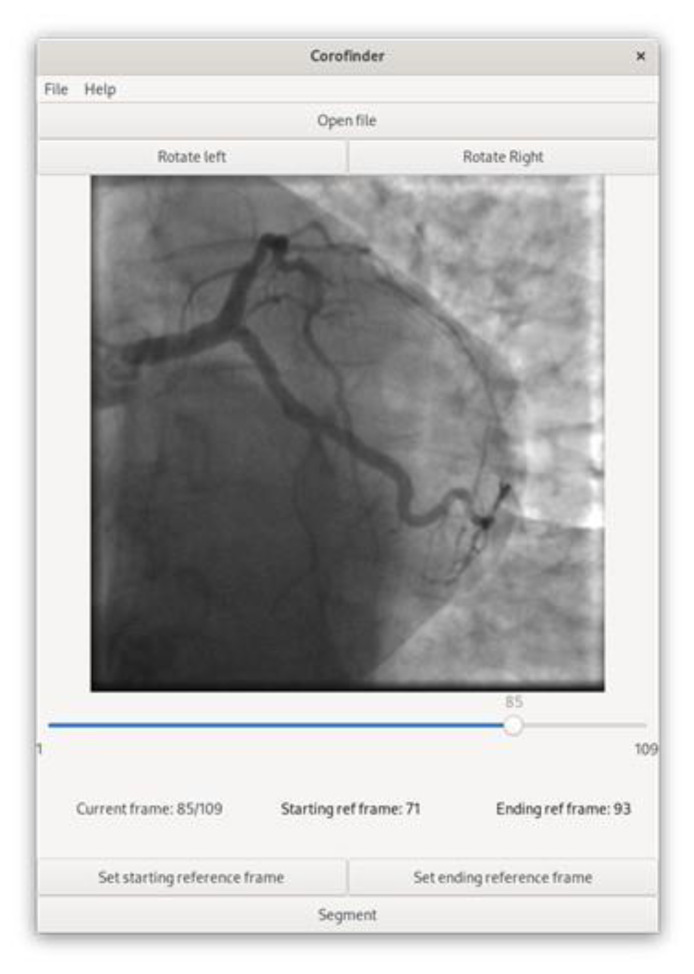
Graphical appearance of the CoroFinder software. In particular, the main page is depicted in the figure.

**Figure 3 jpm-12-00411-f003:**
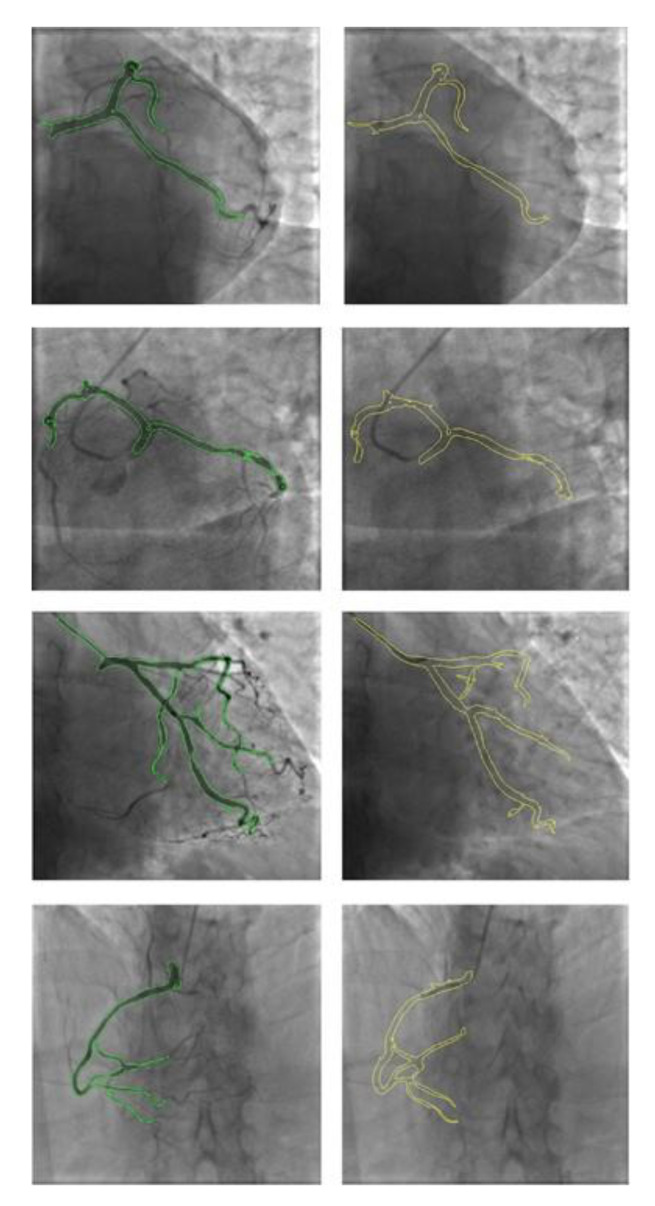
Exemplary frames showing side by side the results of CoroFinder-mediated prediction of vessel position in the right column (the yellow contour indicates the predicted location of the coronary artery) and the segmented contours from standard contrast-enhanced angiograms in the left column (the green contour lines indicate the location of coronary arteries as determined by contrast dye presence). The panel includes three angiographic projections of the left coronary artery: antero-posterior caudal view (first row), left anterior oblique caudal view (second row), right anterior oblique caudal view (third row), and a single angiographic projection of the right coronary artery: left anterior oblique view (fourth row).

## Data Availability

The data presented in this study are available on request from the first author (p.zaffino@unicz.it). The data are not publicly available as CoroFinder is currently under further development to include new additional features.

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
