# Peer review of "CoroFinder: A New Tool for Real Time Detection and Tracking of Coronary Arteries in Contrast-Free Cine-Angiography"

_jpm, 2022, doi:10.3390/jpm12030411_

Round 1

Reviewer 1 Report

In this article, Zaffino and colleagues introduced CoroFinder, a new software to automatically detect and track the profile of epicardial coronary arteries in dye-free images.

This is a proprietary imaging method of coronary arteries. Because of the lack of need to use contrast it is useful and safe for patients with kidney failure.

This presented method may provide direction in imaging diagnostic of coronary arteries.

The article is written correctly. A new imaging technique is clear presented for readers. It shows potential opportunities of use of CoroFinder and the need to continue of research and development of software.

References are up to date and Eanglish is correct.

I recommend this article for fast publication because of innovative nature of the method.

Author Response

Reviewer 1.

In this article, Zaffino and colleagues introduced CoroFinder, a new software to automatically detect and track the profile of epicardial coronary arteries in dye-free images.

This is a proprietary imaging method of coronary arteries. Because of the lack of need to use contrast it is useful and safe for patients with kidney failure.

This presented method may provide direction in imaging diagnostic of coronary arteries.

The article is written correctly. A new imaging technique is clear presented for readers. It shows potential opportunities of use of CoroFinder and the need to continue of research and development of software.

References are up to date and Eanglish is correct.

I recommend this article for fast publication because of innovative nature of the method.

Authors’ answer: we thank the reviewer for taking the time to read our manuscript and the kind words.

Reviewer 2 Report

Dear Authors,

Thank you for your interesting presentation.

Comments:

  1. Please clarify the exact number of pts who underwent coro exam and in how many pts was CoroFinder software used?
  2. Please clarify if your virtual results agreed with classic coronary angiography interpretation. 
  3. Please present the sensitivity and specificity of your method.

Thank you

Author Response

Reviewer 2.

Dear Authors,

Thank you for your interesting presentation.

Comments:

  1. Please clarify the exact number of pts who underwent coro exam and in how many pts was CoroFinder software used?

Authors’ answer: we thank the reviewer for the time and effort spent in revising our manuscript. As stated in the manuscript, the aim of this report was to describe “… the newly developed CoroFinder algorithm and the associated software…” and to provide “… an overview of its application in research and potential for translation to the clinic…”.

Hence, most tests were performed in-silico, while coronary angiogram from two different using multiple projections were used to evaluate its suitability on coronary angiograms from daily routine, as we report in the methods section of the article.

  1. Please clarify if your virtual results agreed with classic coronary angiography interpretation. 

Authors’ answer: we thank the reviewer for this question. As reported in the manuscript, two different approaches were used to verify the correspondence between the coronary contour generated by CoroFinder and coronary angiogram. First, image overlay was exploited to directly compare the vessel contour reconstructed by CoroFinder to the angiographic image. The second step consisted of visual assessment by two independent board-certified interventional cardiologists. We revised the paragraph in the methods section describing this step, better describe the process.

  1. Please present the sensitivity and specificity of your method.

Authors’ answer: as described above, the aim of this study was to describe the “… CoroFinder algorithm and the associated software…”. As such, our aim was to develop the software and verify the feasibility of its use on clinical images. Hence, we did not report data on diagnostic performance because the limited number of cases would yield non-robust results with the potential to overestimate its actual performance (small study effect). We will test diagnostic performance in future larger studies. In particular, we will run a validation study on a specific diagnostic feature, starting with the ability to identify and measure coronary calcium and using coronary CT as the ground truth.

We added an additional sentence to the discussion to make this point clear to the readers.